# Osteonal Microcracking Pattern: A Potential Vitality Marker in Human Bone Trauma

**DOI:** 10.3390/biology12030399

**Published:** 2023-03-03

**Authors:** Nathalie Schwab, Ignasi Galtés, Michelle Winter-Buchwalder, Marisa Ortega-Sánchez, Xavier Jordana

**Affiliations:** 1Biological Anthropology Unit, Department of Animal Biology, Plant Biology and Ecology, Faculty of Biosciences, Universitat Autònoma de Barcelona, Cerdanyola del Vallès, 08193 Barcelona, Catalonia, Spain; 2Forensic Anthropology Unit, Catalonian Institute of Legal Medicine and Forensic Science (IMLCFC), Ciutat de la Justícia, Gran Via de les Corts Catalanes, 111 Edifci G, 08075 Sabadell, Barcelona, Spain; 3Research Group of Biological Anthropology (GREAB), Biological Anthropology Unit, Department of Animal Biology, Plant Biology and Ecology, Faculty of Biosciences, Universitat Autònoma de Barcelona, Cerdanyola del Vallès, 08193 Barcelona, Catalonia, Spain; 4Legal Medicine Unit, Department of Psychiatry and Legal Medicine, Universitat Autònoma de Barcelona (UAB), Cerdanyola del Vallès, 08193 Barcelona, Catalonia, Spain; 5Department of Molecular Biology, Faculty of Life Sciences, University of Vienna, Dr. Bohrgasse 9, 1030 Vienna, Austria; 6Anatomy and Embryology Unit, Morphological Sciences, Faculty of Medicine, Autonomous Universitat Autònoma de Barcelona (UAB), Cerdanyola del Vallès, 08193 Barcelona, Catalonia, Spain; 7Tissue Repair and Regeneration Laboratory (TR2Lab), Institut de Recerca i Innovació en Ciències de la Vida i de la Salut a la Catalunya Central (IrisCC), Ctra. de Roda, 08500 Vic, Barcelona, Spain

**Keywords:** forensic anthropology, bone fractures, blunt force trauma, fracture timing, perimortem trauma, bone histology, bone histomorphometry, microcracking pattern, osteonal microcracks

## Abstract

**Simple Summary:**

When recovered skeletal remains reveal fractures, one of the main challenges is to determine the timing of the trauma. In particular, fractures that are related to death are of high forensic relevance. The common macroscopic criteria used in forensic anthropology may make it possible to determine whether a fracture occurred in a fresh or in a dry bone. Whilst dry bone fractures clearly occurred after death, fresh bone fractures are not always vital fractures. This is one of the biggest issues for forensic anthropologists. Depending on the period of time and surrounding conditions after death, bones can retain their fresh properties due to the preservation of water and organic components. The aim of this study is to test whether the histological assessment of microcracking patterns assists in determining the vitality of a fresh fracture. As we hypothesized after a previous study, our results further support that vital fractures exhibit a higher ratio of osteonal/interstitial microcracks than non-vital fractures. Moreover, we found that axial bone compression may be used to simulate intra vitam conditions in fracture experiments. Our results show that the osteonal microcracking pattern can potentially be used as a marker of vital trauma, thereby improving the probative value of forensic anthropological investigations.

**Abstract:**

In forensic anthropology, the differential diagnosis between peri- and postmortem bone fractures is mainly based on macroscopic criteria. In contrast, studies focusing on bone histology are very scarce. In a recent publication, we showed that (perimortem) fractures in fresh human bones exhibit a different osteonal microcracking pattern than (postmortem) damage in dry bones. In the current work, we explored whether this osteonal microcracking pattern is distinctive of the vitality of (perimortem) fresh bone fractures. To this end, we compared the number, length and structural distribution of microcracks in vital humeral fractures from forensic autopsy cases with experimentally reproduced, three point-bending fractures in fresh and dry human humeri. Half of the fresh experimental bones were fractured whilst applying axial compression, i.e., attempting to simulate intra vitam conditions more accurately. The results showed a similar osteonal microcracking pattern between vital fractures and experimental fractures of fresh humeri subjected to axial compression. Interestingly, this pattern was significantly different from the one observed in the experimental fractures of fresh humeri without axial compression and dry humeri. This supports our hypothesis that the osteonal microcracking pattern can potentially be used as a marker for vital perimortem trauma, providing a histomorphometric tool for fracture timing.

## 1. Introduction

The task of accurately determining the chronology of trauma is one of the most crucial challenges for the forensic anthropologist [1]. The timing of fractures can be categorized into antemortem, perimortem, and postmortem traumas [1,2,3,4]. Antemortem trauma occurs before death and shows bone remodeling as a sign of the initiated healing process in a living body. Similar to antemortem trauma, perimortem trauma also occurs in fresh or wet bone, when the bone still contains its organic components. Yet, the bone shows no signs of healing. The crux is that in forensic anthropology, the term perimortem trauma is used to describe all injuries occurring to fresh (or wet) bone, despite the fact that somatic death may have already occurred [5]. Postmortem damage, in contrast, is a term used by forensic anthropologists to describe fractures that occur in a dry bone when the tissue has lost most of its organic components. This implies the involvement of taphonomic factors such as geological, biological or (un)intentional human alterations [1,2,3,4,5].

The differential diagnosis between fresh bone fractures (perimortem trauma) and dry bone damage (postmortem or taphonomic alterations) is still an issue which forensic experts have to deal with. This is especially difficult when the bones are fragmented or still exhibit some fresh properties. The current gold standard for timing bone fractures is mainly based on macroscopical criteria such as the color or the smoothness and roughness of the fracture margins [2,4,6,7]. Studies applying microscopical methods to distinguish between perimortem and postmortem bone fractures remain scarce and inconclusive [8]. 

In a recent paper, we presented an innovative histological approach to distinguish between peri- and postmortem long bone trauma [9]. In that research, the microcracking pattern of blunt force trauma was compared in two different sample groups: fresh fractured humeri from real traumatic autopsy cases (perimortem trauma) and experimentally fractured dry human humeri (postmortem trauma). The microcrack (MCK) count, length and their occurrence in relation to the microstructure, osteonal or interstitial area were considered. The outcome of that study revealed that fractures in fresh bones (autopsy specimens) exhibited a distinct microcracking pattern compared to fractures in dry bones. Specifically, the results showed that dry bone trauma is characterized by a high number and length of interstitial microcracks (MCKs) and a low number of osteonal MCKs. In contrast, fresh bone fractures showed a lower number of MCKs, particularly in the interstitial area, but a higher proportion of osteonal MCKs. In that paper, we also noted that the increased number and length of interstitial MCKs can be explained by the elevated brittleness of dry bone compared to fresh bone [10].

More difficult to understand, however, is the large number of osteonal MCKs in fresh fractures bones from autopsy cases compared to dry bones. Moreover, the evidence of a higher proportion of osteonal MCKs in fresh fractured bones raises the question of whether this pattern can be used to shorten the time gap of perimortem trauma in order to distinguish vital from non-vital perimortem fractures. As mentioned, forensic anthropologists use the term perimortem trauma when a fracture occurred when the bone tissue was still ‘fresh’ in terms of its wet properties due to the presence of preserved organic components. Consequently, perimortem trauma may or may not be related to the death event itself, and hence, is not always of forensic relevance [5]. The lack of reliable fracture characteristics in the literature allowing the distinction to be made between vital and non-vital perimortem trauma still causes confusion during trauma interpretation. Hence, the identification of fracture markers providing this information would be of huge forensic anthropological interest. 

The objective of the current study is to provide further insight into the microcracking pattern of perimortem trauma in cortical human long bone. In particular, we aim to explore the meaning of osteonal MCKs. In the above mentioned study by Winter-Buchwalder [9], fresh humeri from autopsy cases were analyzed. However, it remained unclear whether osteonal MCKs are only a potential marker for fresh (perimortem) bone trauma or even more specific for vital perimortem trauma. To address this question, we experimentally reproduced perimortem trauma in two sample groups of fresh humeri from human donors. One group was fractured with additional axial compression. The other group was fractured without this additional loading. Whereas the second group basically just simulates fresh bone trauma, the first group may reveal an intra vitam microcracking patterns according to our hypothesis. In this context, the importance of axial bone compression produced by muscular contraction has already been emphasized, in order to replicate intra vitam conditions during experimental bone fractures [11,12]. Thus, to test our hypothesis, we microscopically analyzed the microcracking pattern in experimental blunt force trauma in human humeri with and without axial loading. Then, the microcracking patterns were compared with the results from our previous study using fractured humeri from traumatic autopsy cases and experimental fractures of dry humeri [9]. Being able to characterize the MCK patterns may allow us to optimize the distinction between perimortem and post-mortem trauma, and most notably, between vital and non-vital perimortem trauma.

## 2. Materials and Methods 

### 2.1. Samples

In this study, a total of 10 fresh human humeri were experimentally fractured. We also considered the microcracking patterns of 10 fractured humeri that had already been analyzed and compared in our previous study [9]. All 20 fractured bones resulted from blunt force trauma and revealed a butterfly fracture of the humeral shaft. The specimens, i.e., five of each, were assigned to four sample groups: (a) experimentally produced fractures in fresh humeri with the application of axial bone loading during the experiment (FHC), (b) experimentally produced fractures in fresh humeri without the application of axial loading (FHNC), (c) humeral fractures from real traumatic autopsy cases (AH) [8] and (d) experimentally produced fractures in dry humeri (DH) [9]. Hereby, the microcracking pattern of the autopsy cases was considered as a reference for vital trauma and the dry bone fractures as a reference for postmortem trauma. For the current study, we created two additional experimental groups consisting of fresh human bones in order to simulate a vital and non-vital perimortem MCK pattern.

The autopsy samples were collected from five traumatic autopsy cases at the Institute of Legal Medicine and Forensic Science of Catalonia (IMLCFC). The cadavers were involved in traffic accidents or falls. Only cases with individuals aged 55 or older at death were included in our study in order to reduce age-related bone differences between the real traumatic cases and the experimental samples. 

The experimental bone samples were obtained in an unfractured condition from deceased adults (>60 years) that had donated their bodies to science. The donor bones were provided by the donation program of the Medical Anatomy Department of the Universitat Autònoma de Barcelona (UAB). Prior to their use, all specimens were visually examined in order to exclude macroscopic pathologies or pre-existing traumatic defects. The postmortem time interval (PMI) was the time between death and the experimental use of the respective bone sample. The PMI of all ten fresh humeri was <24 h and the PMI of all five dry humeri was from 15 to 20 years. The 15 donor bones were used to experimentally reproduce a diaphyseal fracture. The fresh donor bones were defleshed with surgical tools prior use. The periosteum, however, was left intact. The dry bones had already been stored without any soft tissue.

### 2.2. Fracture Reproduction

The experimental fractures of the fresh bones were achieved using a pendulum impact test machine [9]. This construction consists of a metal frame and a pendulum to which a hammer with a weight of 5 kg is mounted [12]. The bones were placed horizontally in the metal frame, whilst the anterior side of the shaft faced the hammer. The ends of the humeri were attached to two movable metal holders with tie-wraps. The hammer hit perpendicularly the shaft causing three-point bending. Direct contact with the bone was avoided by attaching a piece of soft rubber to the hammer surface.

Five fresh humeri were fractured during an additional application of axial loading in order to simulate muscular contraction as an intra vitam condition. This experimental condition has already been used in previous studies that worked under the same assumption [13,14]. The axial loading was produced by the attachment of metallic tensors around the proximal and distal epiphyses. In so doing, we assumed that when the bone fractures, the tensors would cause a simulation of muscle contraction. The other five fresh humeri, as well as the five dry humeri [9], were fractured without this additional loading. 

### 2.3. Bone Preparation and Histological Analysis 

The experimentally reproduced fresh fractures were histologically compared with each other and to the fresh fractures from the autopsy cases and the dry experimental fractures. To this end, the freshly fractured specimens were first cleaned to remove the still attached periosteum. Briefly, the samples were cooked in a water detergent solution at a temperature of 90 to 100 °C for 2 to 5 h. After that, the bones were cleaned with paper and left to dry. 

In order to produce thin sections for the histological examination, the bones were first transversally cut into 4 cm long pieces containing the fractured area using an oscillating saw. Each sample was fixated and dehydrated according to the protocol by Ebacher et al. [15] and de Boer et al. [16]. The bone pieces were sequentially left in a series of ethanol solutions (70%, 80%, 90%, 100%) for 24 h each. After being left to dry, the cut bone samples were embedded in epoxy resin. These bone blocks were then cut 1 cm below the main fracture line, ground, polished and mounted on frosted glass. With a Buehler Isomet saw, 100 μm thick sections were cut from the fixed bone blocks. The thin sections were first put into an alcohol gradient (70%, 96%, 100%) and then into the cleaning agent (Histolemon^®^) in order to fix the section. Subsequently, dibutylphthalate polystyrene xylene (DPX) was applied as a mounting medium to fix the coverslip onto the slide until it was polymerized. This technique has already been used in our previous research [9]. 

The histological examination of the thin sections was performed using a Leica DMD 108 microscope. In this way, we obtained micrographs of the whole cortical circumference at a 4x magnification. The microcracking pattern was further analyzed using ImageJ v.1.51. This tool allowed us to calculate the MCK number and to measure the MCK length (μm). In addition, the occurrence of the MCKs was divided into the two microstructural areas of osteonal and interstitial cortical bone [17]. In so doing, the MCKs could be subdivided in either osteonal or interstitial MCKs. In cases where MCKs spread to both areas, fractures were defined according to the most affected one. Overlapped MCKs were not considered. 

The microcracking pattern was evaluated in several cortical fields with a size of 1 mm^2^. For each field, we counted the total MCK number, as well as the number of osteonal and interstitial MCKs (MCK count). Furthermore, we measured the length of the interstitial MCKs (μm). The length of the osteonal MCKs was not considered, as they are all restricted to the border of the osteons which is defined by the cement line. In addition, we calculated the proportion of osteonal MCKs in relation to the total MCKs (number of osteonal MCKs/number of total MCKs) in each field.

### 2.4. Statistical Analysis

We calculated the mean, median, standard deviation (SD) and interquartile range (IQR) for descriptive statistics. For pairwise comparisons, the non-parametric Kruskal-Wallis test and the Dwass-Steel-Critchlow-Fligner (DSCF) test were applied. All statistical analyses were performed using Jamovi 1.0.4. A confidence interval of 95% was applied where *p*-values lower than 0.05 were considered as statistically significant. 

## 3. Results

Table 1 and Figure 1 show a summary of the statistics of the MCK variables. The MCK pattern in the histological bone sections can be seen in Figure 2 and Figure 3. 

The total MKC count $\left(\bar{x}=18.5\pm 10.8\right)$ was significantly highest in the fractured dry humeri (*p* < 0.001). Humeral fractures from autopsy cases displayed a higher MCK count $\left(\bar{x}=8.1\pm 6.0\right)\;$ than the experimental fractures in fresh humeri, although the difference was only significant compared to the fresh humeri with axial compression (*p* = 0.008). There were no significant differences (*p* = 0.165) between both groups of experimental fractures in fresh humeri, i.e., with axial compression $\left(\bar{x}=5.1\pm 3.5\right)\;$ and without axial compression $\left(\bar{x}=6.1\pm 4.3\right)$.

The highest proportion of osteonal MCK was found in the autopsy samples $\left(\bar{x}=0.44\pm 0.28\right)$. Fractured fresh humeri with axial compression featured a proportion of osteonal MCK $\left(\bar{x}=0.37\pm 0.28\right)$ which was not significantly different from autopsy cases (*p* = 0.134). However, there were significant differences (*p* = 0.034) between the proportion of osteonal MCK in both sample groups of experimental fractures in fresh humeri, with and without compression $\left(\bar{x}=0.28\pm 0.23\right)$. Fractured dry humeri showed the lowest proportion of osteonal MCK $\left(\bar{x}=0.13\pm 0.09\right)$. Differences were statistically significant in comparison with the other sample groups (*p* < 0.01). 

The MCKs were significantly longer $\left(\bar{x}=187\pm 124.5\;µm\right)$ in the dry humeri group (*p* < 0.001). Differences in the MCK length were not statistically significant (*p* > 0.2) between the fresh humeri samples (autopsy humeri: $\bar{x}=169\pm 118.6\;$ µm; fresh humeri with axial compression: $\bar{x}=155\pm 90.7\;$ µm; fresh humeri without axial compression: $\bar{x}=169\pm 125.1\;$ µm). A considerable variability in MCK length was observed in all four sample groups.

## 4. Discussion

To increase the quality of forensic anthropological assessments, more reliable data allowing the timing of bone fractures are needed. In forensic anthropology, the classification of perimortem trauma only reveals that a fracture occurred in a fresh or wet bone, despite the fact that somatic death may have already occurred [5]. Thus, further evidence is not only needed to differ between perimortem and postmortem trauma, but also to shorten the perimortem gap in order to distinguish vital perimortem trauma from non-vital perimortem trauma. To gain such data, experimental research under controlled conditions using real human bones is very valuable. By using fresh bones, we simulated perimortem trauma, and by using dry bones, we simulated postmortem trauma. More challenging, however, is the simulation of additional intra vitam conditions in cadaveric long bones. In this context, some authors have already emphasized the importance of applying axial compression during fracture reproduction [11,12]. These studies mainly reported differences in fracture patterns from a macroscopical point of view. In our study, we assessed whether axial loading influences the microcracking pattern, and hence, can simulate intra vitam conditions from a histological point of view. In so doing, we reproduced experimental fractures in fresh human humeri with and without axial loading and in dry human humeri. The microcracking patterns were compared between the three experiment groups and humeral fractures from real traumatic autopsy cases. 

We determined that the total MCK count and the MCK length allowed us to distinguish between fresh bone (perimortem) and dry bone (postmortem) fractures. Both variables were significantly higher in dry fractured bones than in fresh fractured bones (Figure 2). A potential explanation lies in the brittleness of dry bones that lack water and organic components such as lipids and collagen [10]. This increases the impact sensitivity and provokes fractures at a lower strain resistance compared to fresh or wet bone [1,2,3,4,5]. Axial compression, in contrast, had no significant influence on these two variables. Regarding the MCK length, we observed a great variability within all four sample groups. This was likely a result of the inhomogeneous degree of mineralization and microstructure such as osteonal distribution in cortical bone [18,19,20,21,22]. In the latter context, we found that MCKs often stop when they encounter a cement line (Figure 3). This is in accordance with findings by Boyce et al., who reported that interstitial MCKs spread between cement lines [23]. Our finding that the autopsy samples featured a higher total MCK number and length compared to the experimental fractures on fresh bone may be attributed to different impact energies. As noted, the underlying trauma in the autopsy cases were traffic accidents and falls that most probably transferred more kinetic energy than our pendulum impact test machine. 

In general, we found that the majority of MCKs were located in the interstitial area (Figure 2), irrespective of the bone properties. This finding is in line with the literature, and leads to the assumption that MCK initiation, spread and accumulation is overall easier in the broad and relatively brittle interstitial region [9,17,24]. The higher brittleness of the interstitial area compared to the osteons can be attributed to the accumulation of enlarged hydroxyapatite crystals resulting from a lack of remodeling [24].

Whilst the number of interstitial MCKs was significantly higher in the dry bones, the proportion of osteonal MCKs in relation to the total MCKs was significantly higher in the freshly fractured bones (Figure 3). In our previous work, we found a link between the proportion of osteonal MCK and fresh fractures from autopsy cases [9]. Interestingly, in the current study, we found that the proportion of osteonal MCK in fresh humeri fractured with axial compression was similar to that in autopsy cases, whereas the fresh humeri fractured without compression featured significantly less osteonal MCK. These results support the concept that osteonal MCKs may not only be an indicator for perimortem trauma, but also for vital trauma. 

It is known that dynamic loading leads to a weakening of materials by structural microdamage [25]. Bone is a living material, and thus, is capable of repairing MCKs through bone remodeling, a cellular process in which osteoclasts resorb damaged bone and osteoblasts subsequently produce new tissue [26,27]. It is thought that bone remodeling is induced by mechanosensitive osteocytes following physical impact such as stress or strain [28,29,30,31]. The osteocytes sense the environment and act as transductors while they communicate through a fine net of channels named canaliculi [32]. 

The production of MCKs is considered a mechanism to enhance bone toughness, i.e., allowing it to withstand force [9]. It has been claimed that MCKs result from regular mechanical loading and remodeling is required to avoid macroscopic failure [27]. During bone fracture, it is thought that the production of MCKs limits fracture propagation [10,33]. In this way, MCKs dilate the bone tissue and dissipate the impacted energy, which slows fracture growth [10,34]. Furthermore, the occurrence of MCKs decrease the bone stiffness [33]. In this context, the production of MCKs can be considered as a protection mechanism to limit or even prevent macroscopic bone failure. MCKs interfere with the integrity of osteocytes and can easily be repaired by the induction of the above-mentioned bone remodeling [10]. The density of osteocytes is higher in the osteons compared to the interstitial area [35]. Thus, the formation of MCKs in osteocyte rich osteons seems reasonable. However, the mechanical mechanism behind this, i.e., leading to significantly more formation of osteonal MCKs in fresh bones compared to dry bones, remains unclear and demands more research.

A further question arising from our results is the nature and extent of the influence of axial loading on the formation of osteonal MCKs. In the literature, void structures, such as Haversian canals and canaliculi, have been reported to provide stress concentration sites for crack initiation [5,15,31,36,37,38]. Ebacher et al. reported that transversal loading resulted in stress accumulation and MCK initiation around the canaliculi [15]. In comparison, axial loading did not have the same effect on the canaliculi, but the authors suggested that it might have a similar effect on the longitudinally oriented Haversian canals. Consequently, we hypothesize that axial loading propagates along Haversian canals, and hence, increases the stress and strain concentration in the surrounding bone lamellas. This probably facilitates osteonal MCK formation. This assumption is further supported by our observation that osteonal MCKs typically start at the Haversian canal and stop at the cement line (Figure 3A). From another perspective, osteons can be considered to indicate the area where microdamage is most likely to occur. As a self-organizing mechanobiological process, osteons form as a result of microdamage and therefore align along the principal stress [25,39,40]. This mechanism may also be considered as an advantage, because it allows the bone grain to reorient in order to improve its mechanical effectiveness [33].

### Limitations of the Study

We consider the small sample size in each category as a major limitation of this study. Defleshing the bones by macerating them in hot water is a widely used method in forensic anthropology [12,13]. The effect of this method on bone microcracking was not analyzed in the current study. Further research should be done to investigate the potential effects of this process. The fact that there was a definite pattern in the microcracking, and the presence of significant differences between the sample groups, does not give rise to the assumption of such a taphonomic influence. Moreover, this study did not consider other potential important variables such as age and sex on bone mechanical properties. This research is an exploratory approach, and thus, caution is advised when applying these results in routine forensic anthropological fieldwork. 

## 5. Conclusions

The results of this exploratory study reveal histological differences regarding MCK patterns between peri- and postmortem bone trauma. In particular, they indicate that osteonal MCKs are not only associated with perimortem trauma but also with vitality during fracture. This leads us to conclude that osteonal microcracking may be a potential vitality marker in forensic anthropological fracture timing. To simulate intra vitam conditions in trauma experiments, this study supports the application of axial compression. Nevertheless, further research is needed to accurately define the osteonal MCK pattern as a vitality marker and to better understand the mechanism behind MCK formation in general. 

## Figures and Tables

**Figure 1 biology-12-00399-f001:**
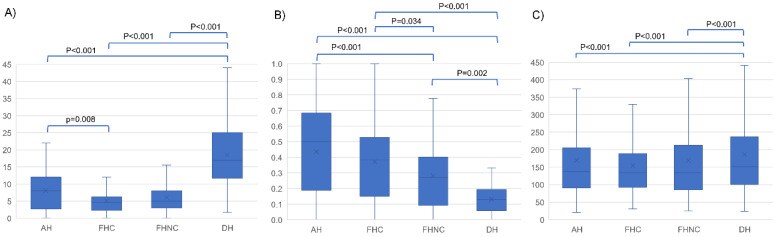
Boxplots of the MCK variables by the samples. (**A**) MCK count. (**B**) Proportion of osteonal MCKs. (**C**) MCK length (µm). Significant *p*-values from DSCF test for pairwise comparisons are shown. AH (autopsy humeri); FHC (fresh humeri with compression); FHNC (fresh humeri without compression); DH (dry humeri).

**Figure 2 biology-12-00399-f002:**
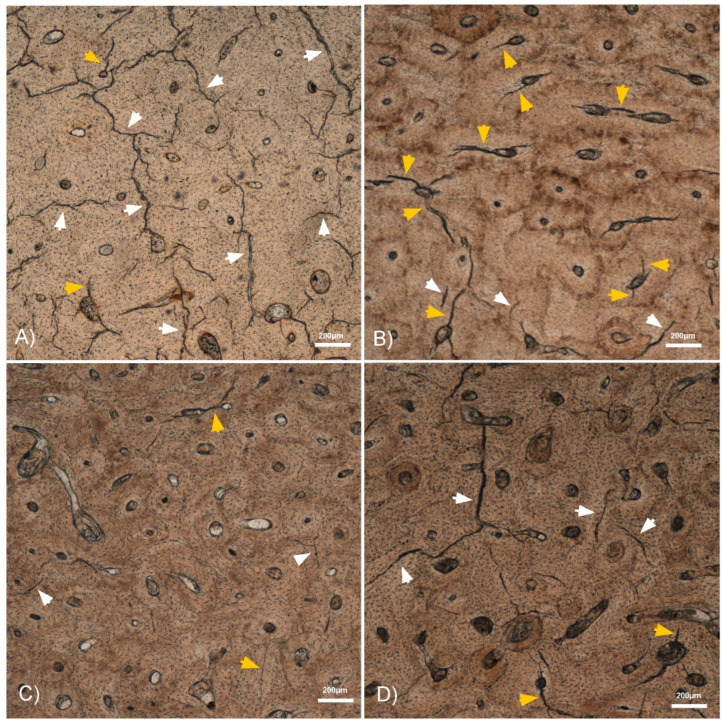
4× micrographs of the histological bone sections showing some interstitial (white arrowheads) and osteonal (orange arrowheads) MCKs. (**A**) Dry humerus with a high number of long interstitial MCKs. (**B**) Autopsy humerus with a high number of osteonal MCKs. (**C**) Fresh humerus with compression showing both osteonal and interstitial MCKs. (**D**) Fresh humerus without compression showing more interstitial than osteonal MCKs.

**Figure 3 biology-12-00399-f003:**
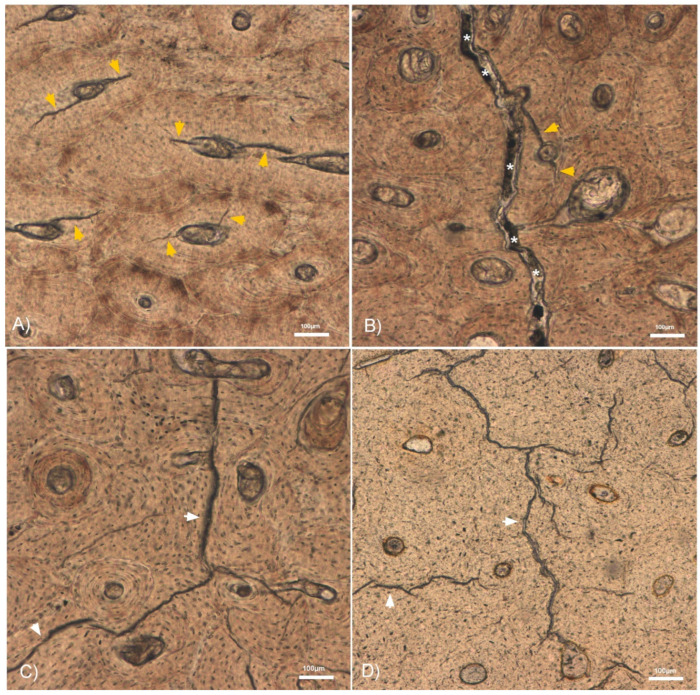
10× micrographs of the histological bone sections showing some interstitial (white arrowheads) and osteonal (orange arrowheads) MCKs. (**A**) Autopsy humerus with osteonal MCKs running from the Haversian canal to the cement line. (**B**) An osteonal MCK and the fracture line (*) in a fresh humerus with compression. (**C**) A long interstitial MCK in a fresh humerus without compression. (**D**) Dry humerus with long interstitial MCKs.

**Table 1 biology-12-00399-t001:** Descriptive statistics of the MCK variables by the samples. *n* the number of areas analyzed (MCK count, Proportion of osteonal MCKs), or the number of MCKs measured (Interstitial MCK length).

	Sample Groups	*n*	Mean	Median	SD	IQR
MCK count	AH	64	8.10	8.00	6.01	9.31
	FHC	74	5.07	4.63	3.48	3.94
	FHNC	82	6.09	5.00	4.30	5.00
	DH	29	18.53	17.00	10.76	13.25
Proportion of osteonal MCKs	AH	64	0.44	0.50	0.28	0.47
	FHC	74	0.37	0.39	0.28	0.35
	FHNC	82	0.28	0.27	0.23	0.31
	DH	29	0.13	0.13	0.09	0.11
Interstitial MCK lenght (µm)	AH	922	169.00	137.00	118.60	113.30
	FHC	408	155.00	135.00	90.70	95.60
	FHNC	945	169.00	134.00	125.10	126.90
	DH	1304	187.00	151.00	124.50	136.20

AH (autopsy humeri); FHC (fresh humeri with compression); FHNC (fresh humeri without compression); DH (dry humeri).

## Data Availability

The data generated during and/or analyzed during the current study are available from the corresponding author on reasonable request. All samples are stored in a private collection at the IMLCFC (Barcelona, Spain), registered as a collection at the Instituto de Salud Carlos III (Reference C.0004241).

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
