# Peer review of "Osteonal Microcracking Pattern: A Potential Vitality Marker in Human Bone Trauma"

_biology, 2023, doi:10.3390/biology12030399_

Round 1
Reviewer 1 Report
In this article, the authors focused on osteonal microcracking pattern as potential marker for the vitality of fresh bone fractures. To this purpose, experimental perimortem traumas were performed on fresh bone samples simulating intra vitam conditions and the microcracking pattern was compared among samples and with previous data on real traumatic autopsy cases and postmortem fractures on dry bones.
Determining the vitality of a bone fracture is one of the most difficult challenges in forensic anthropology and few data is available so far on vital microscopic features. This study is of great interest and well presented.
I consider the article valuable to be published in this journal. However, before being published, the authors could consider a minor revision.
- P. 2, line 64, the word "MCK" appears for the first time, the authors could report to this line the meaning in lines 68-69
- P. 3, line 134, "six dry humeri", please check if six is the corret number
- A similar study was performed in 2011 (Pechníková, M., Porta, D., & Cattaneo, C. (2011). Distinguishing between perimortem and postmortem fractures: are osteons of any help?. International journal of legal medicine, 125, 591-595.), even if with some differences. This could be added to the references
Author Response
We would like to thank the reviewer for their comments on our manuscript. Below is the reply detailing the changes in the revised version.
- P2, line 64, the word "MCK" appears for the first time, the authors could report to this line the meaning in lines 68-69
Changed
- P3, line 134, "six dry humeri", please check if six is the corret number
Changed to “five dry humeri”
- A similar study was performed in 2011 (Pechníková, M., Porta, D., & Cattaneo, C. (2011). Distinguishing between perimortem and postmortem fractures: are osteons of any help?. International journal of legal medicine, 125, 591-595.), even if with some differences. This could be added to the references
Reference added (see page 2)
Reviewer 2 Report
The article is well written and the research clearly presented. However, there are some issues with the terminology of peri / postmortem and fresh / wet bone. To me, it was quite confusing. I got the feeling that the terms were not always used properly, neither were the terms properly explained.
Also, using the bones of older individuals (of unknown sex) should be discussed as sex and age can affect bone density. Similarly, there should be some kind of recognition/discussion with the effect of cooking.
Black arrows are not well visible on some of the figures.
Authors clearly presented the limitation of a small sample size.
Author Response
We would like to thank the reviewer for their comments on our manuscript. Below is the reply detailing the changes in the revised version.
- The article is well written and the research clearly presented. However, there are some issues with the terminology of peri / postmortem and fresh / wet bone. To me, it was quite confusing. I got the feeling that the terms were not always used properly, neither were the terms properly explained.
These terms have been defined in the introduction section and references have been cited accordingly. Terminology has been checked along the manuscript.
- Also, using the bones of older individuals (of unknown sex) should be discussed as sex and age can affect bone density. Similarly, there should be some kind of recognition/discussion with the effect of cooking.
We added new paragraph in the limitation section: “Defleshing the bones by macerating them in hot water is a widelly used method in forensic antrhopology [12, 13]. The effect of this method on bone microcracking was not analysed in the current study. Further research should be done to investigate the potential effects of this variable. The fact that there is a definite pattern in the microcraking, and the presence of significant differences between the sample groups does not give rise to the assumption of such taphonomic influence. Moreover, this study does not consider other potential important variables such as age and sex on bone mechanical properties”
- Black arrows are not well visible on some of the figures.
Changed